# Irrational Complex Rotations Empower Low-bit Optimizers

**Zhen Tian**
ByteDance
Beijing
chenyuwuxinn@gmail.com

**Wayne Xin Zhao**[*]
GSAI, Renmin University of China
Beijing
batmanfly@gmail.com

**Ji-Rong Wen**
GSAI, Renmin University of China
Beijing
jrwen@ruc.edu.cn

## Abstract

In this paper, we propose a novel optimizer state compression algorithm, namely $\pi$-**Quant**, which leverages the properties of irrational numbers (*e.g.,* $\pi$) for memory-efficient training. The core idea is based on our mathematical findings, which show that a pair of parameters can be represented by a single rotation angle using the complex rotation scheme. Building on this insight, we map the parameters into a complex space and perform quantization using the corresponding rotation angles. To efficiently integrate it into optimization process, we develop an efficient system of geometric equations that computes the precise rotation angles with linear complexity. We evaluate $\pi$-Quant on a wide range of tasks. Our experiments show that it can reduce the bit-width of parameters to 3.32-bit, achieving a 41.8% decrease in GPU memory usage, all while maintaining full accuracy. The code is provided at `https://github.com/RUCAIBox/Pi-Quant`.

## 1 Introduction

The rapid growth of large-scale AI models (*e.g.,* large language models [1]) has led to significant improvements in various tasks. However, larger models often require substantially more computational power and memory resources to train. Typically, most models are trained with momentum-based optimizers such as Adam [2], which require the storage of one or two optimizer momentum for each individual parameter. As the parameter scale increases, the optimizer states consume the majority of memory, with first-order and second-order momentum tensors in the Adam accounting for 66% of parameter usage [3]. Therefore, finding a solution to develop a memory-efficient optimizer is a critical challenge.

Considering the above issues, a number of studies [4, 3] propose using lower-precision representations for the optimizer states, as shown in Table 1. They primarily analyze the frequently occurring values of the optimizer states, and then use binary search to quantize all state elements to these values. While previous attempts have demonstrated the promise of this approach, two major challenges remain. First, most deep learning frameworks (*e.g.,* Pytorch [5], TensorFlow [6]) do not support such search quantization operations, these methods require additional compilation of GPU-supported operators. As a result, these methods are not directly compatible with devices other than GPUs (*e.g.,* CPU or TPU). Second, these methods are specifically designed for certain bit-width quantization, making it

---

[*]Wayne Xin Zhao (batmanfly@gmail.com) is the corresponding author.

difficult to adapt them to other bit-widths. Since these predefined precisions may not be well-suited to all tasks, they can lead to performance degradation in certain cases.

Table 1: Comparing different quantization methods, where $n$ is the parameter size, and $m$ is the precision size.

| Metric | Bnb (2021) | Lpmm (2023) | $\pi$-Quant (ours) |
|---|---|---|---|
| Bit-width | 8 | 4 | 3.32 |
| Complexity | $O(n \log m)$ | $O(n \log m)$ | $O(n)$ |
| Full Acc. | ✔ | ✘ | ✔ |

To address these issues, we aim to develop a *precise* and *theoretically-guaranteed* compression algorithm that can flexibly reduce the memory usage of optimizer states within a typical model optimization framework. Our key idea is grounded in an important mathematical property: $\forall \{(x,y)|x^2 + y^2 \leq 4\}, \exists \theta \in \mathbb{R}, x + iy \rightarrow e^{i\theta} + e^{i\pi\theta}$. This enables the *precise* representation of a parameter pair $(x,y)$ with a single complex rotation angle (*i.e.,* $\theta$), by harnessing the properties of irrational numbers (*i.e.,* $\pi$). As a result, the parameter scale is halved with no loss of precision. Further, we can quantize the rotation angles to achieve even greater memory reduction. To implement our idea, a fundamental challenge is integrating the proposed complex rotation scheme into the optimization process. Since the optimizers are often designed for real-valued parameters, they are not directly compatible with our complex transformation. Further, it is also difficult to find an efficient solution for accurately computing the rotation angles of the corresponding parameters during optimization.

Motivated by our mathematical findings, in this paper, we propose a novel optimizer state compression approach with irrational complex rotation, named $\pi$-**Quant**. The major contribution of $\pi$-Quant is the introduction of a precise theoretical framework for parameter compression, resulting in lower memory costs during model training. Specifically, $\pi$-Quant involves two key techniques. First, it develops a new representation mechanism that maps the parameters in a complex space. On this basis, the complex-valued parameters are transformed into single rotation angles according to our proposed mathematical formula, which precisely halves the parameter scale. Notably, it employs an efficient system of geometric equations that can compute the precise rotation angles with linear complexity. Second, it features an effective quantization algorithm that reduces the precision of the rotation angles during training, all while preserving full accuracy. Our theoretical framework is particularly robust for numerically sensitive parameters, offering lower quantization errors compared to prior methods. Generally, $\pi$-Quant can be seamlessly integrated into existing optimization pipeline, which largely reduces the training memory footprint, with minimal impact on the training speed.

The main contributions are summarized as follows:

• We propose an effective optimizer state compression approach, named $\pi$-Quant, by introducing a new mathematical formula. This method transforms a parameter pair into a univariate rotation equation with irrational coefficients, enabling precise compression by halving the parameter scale. To the best of our knowledge, we are the first to leverage the properties of irrational numbers for model compression.

• We propose an efficient system of geometric equations for optimizing the compression process, which is capable of computing the precise rotation angle with linear complexity. Additionally, we develop an effective quantization algorithm that reduces the representation precision of the rotation angles, while maintaining full accuracy.

• Experimental results show that $\pi$-Quant is an effective quantizer, capable of reducing the bit-width of optimizer states to 3.32 bits, *e.g.,* it can reduce the training memory of TinyLlama from 19.47 G to 11.32 G, with comparable accuracy. Besides, it consistently outperforms several state-of-the-art quantization methods on a wide range of tasks, highlighting the effectiveness of our approach.

## 2 Preliminaries

In this section, we give a brief introduction to the quantization techniques, and then describe the complex rotation technique used in our approach.

**Training-Oriented Quantization**. Quantization is a compression technique that maps high-precision values to a finite set of discrete values. Formally, given the quantization range $D$, quantization

approaches typically design a mapping function $[0, 2^k − 1] \mapsto D$, where $k$ represents the bit-width of the quantization. The quantization methods can be generally divided into *inference-oriented* and *training-oriented* according to their application scenarios. Typically, inference-oriented methods often conduct post-training quantization of the backbone network (*e.g.,* dense layers) for reducing inference costs. However, these methods are not practical to be applied into training pipeline, since they fail to effectively fit the non-uniform distributed parameters, they often suffer from significant accuracy loss during optimization.

In contrast, training-oriented methods [7, 3] focus on reducing the precision of optimizer states to achieve lower training memory. For better fitting the non-uniform parameter distributions, their basic idea is to sample high-frequency elements to construct a non-uniform quantization range $D$, *e.g.,* $D = \{0.12, 0.25, 0.5, 1.0\}$. Due to such non-uniformity, these methods cannot directly compute the quantization indices and typically require a search-based method to perform the quantization, *i.e.,* $\mathbf{X}_i^{INT} = \arg\min_{\theta \in D} ||\theta − \mathbf{X}_i^{FP32}||$. In practice, these methods often require compiling additional GPU kernels to perform parallel searching. Besides, the certain bit-width setting of $D$ cannot be extended to other bit-widths. As a training-oriented method, our approach focus on reducing the memory of optimizers, which pushes the lowest achievable bit-width (4-bit) down to 3.32-bit, with higher accuracy and no need for a search process.

**Complex Rotation**. In mathematics, complex rotation is a mathematical operation that applies a rotation in the complex plane. It is represented by multiplying a complex number of the form *i.e.,* $\cos\theta + i\sin\theta$, where $i$ is the imaginary unit ($i^2 = −1$) and $\theta$ is the rotation angle. According to Euler's formula, a complex number can be written as:

$$\cos\theta + i\sin\theta = e^{i\theta}. \tag{1}$$

This equation establishes a mapping from $\{(x, y)|x^2 + y^2 = 1\}$ to $\theta \in \mathbb{R}$. In our paper, we reveal a more profound theorem, which shows that arbitrary pair of $(x, y)$ can be represented by an angle $\theta$ in this complex rotation scheme. As such, it provides a new solution to compress the parameters.

## 3 Methodology

In this section, we introduce the proposed $\pi$-Quant method (illustrated in Figure 1) for achieving memory-efficient parameter optimization. Our method is grounded in a key mathematical property: each real vector can be split into two components, which can then be uniquely represented by a rotation angle. Building on this idea, we propose a geometric approach for performing this transformation, enabling the computation of the rotation angle with linear complexity. Moreover, we design a quantization pipeline to implement our method, which significantly reduces the parameter scale and can be seamlessly integrated into existing optimization algorithms. We begin by introducing the theoretical framework of $\pi$-Quant, followed by a description of its practical application pipeline.

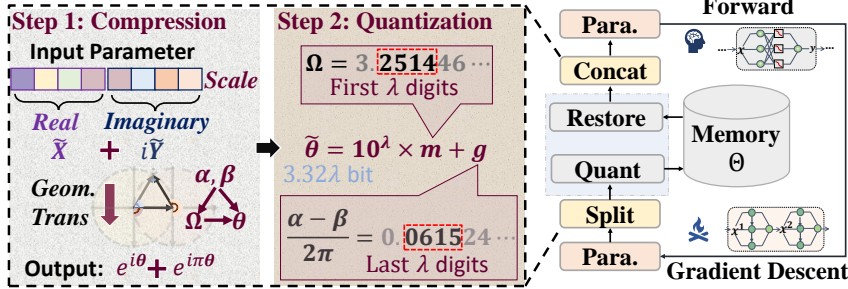

Figure 1: The overall framework of $\pi$-Quant.

### 3.1 Parameter Compression with Irrational Rotations

The key point of our approach is a novel data compression theorem, which enables the precise representation of two variables using a univariate rotation equation with an irrational number as

the coefficient. We first present the mathematical foundation behind this theorem, followed by the instantiation of our idea.

**Data Compression with Irrational Numbers.** Existing machine learning frameworks typically represent model parameters using floating-point numbers (*e.g.,* FP32) with limited precision. In this representation, previous work [8] often employs truncation-like methods to uniformly quantize high-precision parameters to a lower precision. The underlying assumption is that the parameters follow a uniform distribution within the quantization range. However, due to the mismatch between this assumption and the actual distribution of the parameters, most quantization methods encounter significant errors, as parameters are more likely to follow a non-uniform distribution (*e.g.,* Gaussian distribution). In contrast to prior approaches, we introduce a novel data compression technique that compresses parameters from the perspective of irrational numbers. Formally, we present the following theorem:

**Theorem 3.1.** *Given a complex number $z = x + iy \in \mathbb{C}$ that satisfies $\|z\| \leq 2$. For any deviation $\epsilon > 0$, there exists an angle $\theta \in \mathbb{R}$, such that*

$$\left\| z - \left( e^{i\theta} + e^{i\bar{\pi}\theta} \right) \right\| \leq \epsilon. \tag{2}$$

*where $i$ is the imaginary unit, and $\bar{\pi}$ denotes any irrational number (e.g., the circle ratio $\pi$).*

Specifically, it shows that any vector of two real values $x$ and $y$ can be represented by a real-valued angle $\theta$ in this complex scheme. This property extends to any parameter tensor $\mathbf{T}$, which can be split and linearized into two equal-sized matrices, *i.e.,* $(\mathbf{X}, \mathbf{Y}) = \text{Split}(\mathbf{T})$, corresponding to the real part and imaginary part, respectively. In this manner, we can represent $\mathbf{T}$ by a specific angle vector $\mathbf{\Theta}$, using half the number of real values. A proof of this theorem is provided in Appendix A.

**Geometric Solutions.** Finding the exact solution to Eq. (2) is the key challenge of our approach. Intuitively, one might consider using a search-based method (*e.g.,* KDTree), where the $x$ and $y$ values corresponding to all $\theta$ angles within the desired precision are stored in a cached array, and the array is then searched for the coordinates closest to the target $(x, y)$. However, this approach often incurs $O(n \log m)$ complexity and presents challenges in performing parallel searches on a GPU. As shown in Figure 1, to efficiently solve the Eq. (2), we propose a geometric solution that enables faster implementation:

**Lemma 3.2.** *Given the input complex number $x + iy$, the solution $\theta$ of Eq. (2) can be given by the following system of geometric equations:*

$$\begin{cases} \alpha = \arctan(y/x), \\ \beta = \arccos(\sqrt{x^2 + y^2}/2) \\ \theta = \alpha - \beta + 2m\pi, \ m \in \mathbb{Z} \\ \bar{\pi}\theta = \alpha + \beta + 2k\pi, \ k \in \mathbb{Z} \end{cases} \tag{3}$$

We prove this lemma in Appendix B. As such, the task of finding the exact value of $\theta$ is transformed into: given $(x, y)$, determine the specific value of the integer $m$, such that there exists an integer $k$ satisfying the above system of equations. By further simplifying the system of Eqs. (3), our main objective becomes solving the following equation:

$$\{m\bar{\pi}\} = \{\Omega\}, \tag{4}$$

$$\Omega = \frac{\alpha(1 - \bar{\pi}) + \beta(1 + \bar{\pi})}{2\pi}, \tag{5}$$

where $\{\cdot\}$ returns the fractional part of a input number. Here, we can first compute the exact value of $\Omega$ with linear complexity. However, directly solving Eq. (4) is very difficult. Intuitively, we can enumerate all possible values of $m$ and then search for the longest match with the fractional part of the target $\Omega$. Nevertheless, the time complexity of the search process is still extremely high and not practical. As our solution, we selectively mask certain digits of $\pi$. Specifically, the value of $\bar{\pi}$ in our approach is constructed as:

$$\bar{\pi} = 0. \underbrace{000\cdots}_{\lambda-1 \text{ times}} 1 \underbrace{0000\cdots}_{\lambda \text{ times}} \underbrace{3589793238\cdots}_{\text{same as } \pi}, \tag{6}$$

where $\lambda$ is a quantization hyper-parameter. We can guarantee that $\bar{\pi}$ is still an irrational number, as it can be obtained by adding $\pi$ to some rational numbers. In this way, we can approximate $m$ to the first $\lambda$ decimal places of $\Omega$:

$$m \approx \lfloor \{\Omega\} \times 10^\lambda \rfloor \tag{7}$$

where $\lfloor \cdot \rfloor$ represents the floor function. We give an intuitive explanation of our construction in Appendix E. This solution guarantees that the error does not exceed $10^{-\lambda}$. The entire process maintains linear complexity.

## 3.2 Optimization with Rotation Quantization

In the previous section, we discussed how to compress a pair of real-value parameters using a single rotation variable $\theta$. To further reduce the memory usage, we aim to use a low-precise representation for quantizing the corresponding $\theta$. Next, we will introduce the details of our framework.

**Quantize the Rotation Angles.** As introduced in Section 3.1, our approach use the complex rotation scheme to represent parameter pairs. One limitation of this method is that it requires the magnitudes of $x$ and $y$ (*i.e.*, $||x^2 + y^2||$) to be no more than 4. To address this, we first scale and normalize the parameters to ensure they lie within the representable range. Formally, given the split input matrices $(\boldsymbol{X}, \boldsymbol{Y}) = \text{Split}(\mathbf{T})$, the scaled tensor can be computed by a simple method:

$$\tilde{\boldsymbol{X}} = \boldsymbol{X}/w, \ \tilde{\boldsymbol{Y}} = \boldsymbol{Y}/w \tag{8}$$

where $w$ is the maximum absolute value of the numbers in both $\boldsymbol{X}$ and $\boldsymbol{Y}$. Using this method, the value ranges of $\tilde{\boldsymbol{X}}$ and $\tilde{\boldsymbol{Y}}$ are both transformed to $[-1, 1]$, satisfying the mapping conditions in Theorem 3.1. Note that we can also constrain them to a larger range (*e.g.*, $[-\sqrt{2}, \sqrt{2}]$) by multiplying $w$ by a coefficient. Empirically, this setting has already achieved a good trade-off in quantization deviation, which we will discuss in Section 3.3. Afterwards, we compress the $\tilde{\boldsymbol{X}}$ and $\tilde{\boldsymbol{Y}}$ to a unique $\Theta$, following the method introduced in Section 3.1. For further quantizing $\theta$, we retain $\lambda$ decimal places of $\alpha - \beta$ (See Eq. (3)), with $10^{-\lambda}$ as the interval:

$$\begin{aligned} \tilde{\theta} &= \frac{\theta}{2\pi 10^{-\lambda}} = m \cdot 10^\lambda + g, \\ g &= \lfloor \frac{\alpha - \beta}{2\pi} \times 10^\lambda \rfloor \end{aligned} \tag{9}$$

In this way, we can represent each $\theta$ with $2\lambda$ digits, where the first $\lambda$ digits correspond to the $m$ and the remaining $\lambda$ digits correspond to the $g$. As such, we can set varied $\lambda$ to achieve different bit-widths. Since the $\lambda$ is set based on the decimal system, on average, each parameter requires $\lambda \cdot \log_2 10 \approx 3.32\lambda$ bits for storage. In particular, the entire process remains linear complexity and supports parallel computation on GPUs, making it practical in the large scale model optimization. We present a detailed procedure for the quantization process in Algorithm 1.

Through the rotation transformation (Eq. (2)), the parameter scale is reduced by half. After quantization, the precision of the angle representation can be further reduced. In practice, we can compress the precision of an angle into an integer ranging from 0 to 99, achieving the same quantization effect as 3.32-bit (See Table 3). Next we discuss how to apply our algorithm during the training process.

**Optimization.** The main advantage of our algorithm lies in its ability to reduce the optimization memory. Since only a small part of parameters necessitate immediate computation during training [3], we only restore this subset of parameters, while the remainder continues to reside in memory in their angular representation. As shown in Figure 1, in our framework, the parameters stored in memory are $\Theta$ and the corresponding $w$ (See Algorithm 1). When the parameters are involved in computing, we apply the inverse operations of the quantization process to activate them:

$$\begin{cases} \theta = \tilde{\theta} \times 2\pi 10^{-\lambda}, \\ \tilde{x} = \cos\theta + \cos\bar{\pi}\theta, \ \tilde{y} = \sin\theta + \sin\bar{\pi}\theta, \\ x = \tilde{x} \cdot w, \ y = \tilde{y} \cdot w. \end{cases} \tag{10}$$

**Algorithm 1:** Quantization of the Rotation Angles

1: **Input:** Parameter $\mathbf{T}$, size: $n$
2: Split $\mathbf{T}$ into two tensors: $\boldsymbol{X}$ and $\boldsymbol{Y}$
3: Compute $w = \max(|\boldsymbol{X}, \boldsymbol{Y}|)$
4: Scale $\boldsymbol{X}$ and $\boldsymbol{Y}$ according to Eq. (8)
5: Compute $\boldsymbol{\alpha}$ and $\boldsymbol{\beta}$ using Eq. (3)
6: Compute $\boldsymbol{\Omega}$ according to Eq. (4)
7: Compute $\boldsymbol{m}$ based on Eq. (7)
8: Calculate $\boldsymbol{\Theta}$ from Eq. (9)
9: **Output:** Quantized parameter $\boldsymbol{\Theta}$, size: $n/2$; Scale Factor $w$, size: 1

**Algorithm 2:** Adam with $\pi$-Quant (Differences highlighted)

1: **Input:** learn rate $\alpha$, decay rates $\beta_1, \beta_2, \epsilon$
2: **Initialize:** $\boldsymbol{m}_0 = \text{Quant}(\mathbf{0})$, $\boldsymbol{v}_0 = \text{Quant}(\mathbf{0})$, $t = 0$ {using Alg. 1}
3: **for** each iteration $t = 1, 2, \ldots, T$ **do**
4:     Compute gradient $\nabla \boldsymbol{\theta}_t$
5:     $\boldsymbol{m}_{t-1} \leftarrow \text{Restore}(\boldsymbol{m}_{t-1})$ {Eq. (10)}
6:     $\boldsymbol{m}_t \leftarrow \beta_1 \boldsymbol{m}_{t-1} + (1 - \beta_1) \nabla \boldsymbol{\theta}_t$
7:     $\boldsymbol{v}_{t-1} \leftarrow \text{Restore}(\boldsymbol{v}_{t-1})$ {Eq. (10)}
8:     $\boldsymbol{v}_t \leftarrow \beta_2 \boldsymbol{v}_{t-1} + (1 - \beta_2)(\nabla \boldsymbol{\theta}_t)^2$
9:     $\hat{\boldsymbol{m}}_t \leftarrow \frac{\boldsymbol{m}_t}{1 - \beta_1^t}, \hat{\boldsymbol{v}}_t \leftarrow \frac{\boldsymbol{v}_t}{1 - \beta_2^t}$
10:     $\boldsymbol{\theta}_t \leftarrow \boldsymbol{\theta}_{t-1} - \alpha \frac{\hat{\boldsymbol{m}}_t}{\sqrt{\hat{\boldsymbol{v}}_t} + \epsilon}$
11:     $\boldsymbol{m}_t \leftarrow \text{Quant}(\boldsymbol{m}_t)$ {Alg. 1}
12:     $\boldsymbol{v}_t \leftarrow \text{Quant}(\boldsymbol{v}_t)$ {Alg. 1}
13: **end for**
14: **Output:** Optimized parameters $\boldsymbol{\theta}_T$

In this way, we can easily recover the parameter tensor $\boldsymbol{X}, \boldsymbol{Y}$ before quantization, and we then concatenate them to reconstruct the original $\mathbf{T} = \text{Concat}(\boldsymbol{X}, \boldsymbol{Y})$. Similar to existing quantization works [3, 7], the activated parameters maintain high precision to ensure the accuracy of the gradients and predictions. Since only a small subset of parameters is involved in computation most of the time, the temporary storage overhead incurred by these high-precision activated parameters is negligible. The detailed algorithm is presented in Algorithm 2.

Like most training-oriented quantization methods [7, 3], $\pi$-Quant introduces additional "Quant" and "Restore" computations for each iteration, which results in higher latency compared to the original Adam. Specifically, given the parameter size $n$, the additional complexity of $\pi$-Quant is $O(n)$ per iteration, which can be negligible compared to the overall time complexity.

### 3.3 Discussion

In this part, we discuss the advantage of $\pi$-Quant and compare our work with existing work.

**Model Merits.** Our method has the following merits:

• *Quantization Precision.* We analyze the quantization accuracy in $\pi$-Quant, and demonstrate its superiority over traditional approaches. Specifically, the main deviation of our method comes from the approximation of the $m$ and $g$ in Eq. (7) and Eq. (9), respectively. Formally, given the deviation of $\Delta\theta$ in Eq. (9), the upper bound of quantization error for a specific point $(x, y)$ is given by:

$$
\begin{aligned}
||\Delta x|| &< \Delta\theta \Big|\Big|(1 - \bar{\pi})\sin\theta' + \bar{\pi}y'\Big|\Big| < \Delta\theta, \\
||\Delta y|| &< \Delta\theta \Big|\Big|(1 - \bar{\pi})\cos\theta' + \bar{\pi}x'\Big|\Big| < \Delta\theta,
\end{aligned}
\tag{11}
$$

where $\theta' \in [\theta, \theta + \Delta\theta]$. We give a proof in Appendix C. It indicates that our method can reduce the quantization error in the corresponding angle $\theta$. Further, given the precision width $\lambda$, the average quantization error can be given by:

$$
\mathbb{E}(\Delta x) = \mathbb{E}(\Delta y) = O(2 \cdot (1 + \bar{\pi}) \cdot 10^{-\lambda}/\pi).
\tag{12}
$$

The proof is in Appedix D. This indicates that the quantization error increases as the bit width decreases. Note that $2 \cdot (1 + \bar{\pi}) < \pi$, which further demonstrates that our method has a lower average quantization error compared to traditional methods (*i.e.,* $10^{-\lambda}$). In practice, we reduce the bit-width of optimizer states down to 3.32-bit, while maintaining full accuracy (See Section 4.1).

• *Non-uniformity.* Non-uniformity is an essential property that ensures the accuracy of quantization for optimization, as the parameters of neural networks typically exhibit a non-uniform distribution (*e.g.,* Gaussian distribution). In Figure 2, we plot the quantization error distribution between traditional

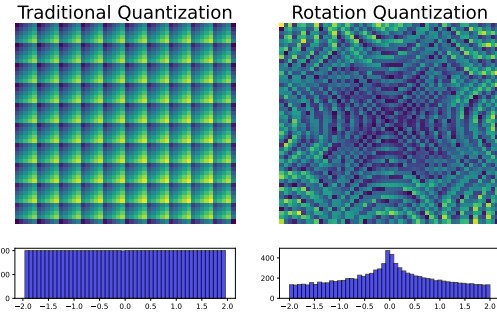

| Methods | Bit-width | Search | Distribution | Unit |
|---------|-----------|--------|--------------|------|
| LinearQuant | Flexible | No | Uniform | Single |
| Bnb.Adam | 8 | Yes | Non-Uniform | Single |
| Lpmm.Adam | 4 | Yes | Non-Uniform | Single |
| $\pi$-Quant (ours) | 3.32 (Flexible) | No | Non-Uniform | Pair |

Table 2: Comparison of different quantization methods.

Figure 2: The error (top) and precision distribution (bottom) between $\pi$-Quant and traditional methods.

quantization methods and our approach. As shown, the errors of traditional methods are uniformly distributed across the parameter space. In contrast, our method produces smaller errors near the zero point, with larger errors at the periphery. Additionally, we plot the accuracy distribution for both traditional methods and our approach. It is evident that our method allocates more precision around the zero point to better capture the parameters, while assigning less precision to the range of larger values. Since most parameters and momentum in large models follow a standard distribution with a mean close to zero, our method is more effective at quantizing these parameters.

**Novelty and Differences.** In Table 2, we compare our method with optimizer state quantization methods. To the best of knowledge, it is the first attempt that leverages the properties of irrational numbers for data compression, grounded in a new mathematical formula. This approach transforms a complex-valued parameter into a single rotation angle, which can precisely halve the parameter scale of a wide range of modules such as optimizer states and model weights. By the proposed system of geometric equations, our approach only requires linear-complexity calculations, possessing superior theoretical properties (*e.g.,* non-uniformity), with no need of complicated search process. In contrast to traditional quantization methods [3, 7], which perform numerical compression for each single element, we propose to conduct the quantization for each pair unit of parameters, by reducing the precision of the rotation angles. As such, it provides a way to better capture the correlations between different elements, making it possible to further reduce precision to lower bit-widths. In practice, $\pi$-Quant can reduce the precision of optimizer states to 3.32-bit, which can be flexibly extended into variable bit-widths, with minimal impact on the training speed (See Table 3). In general, our method provides a precise and theoretically-grounded solution for compressing the parameter scale for optimization.

## 4  Experiment

### 4.1  Language Modeling Evaluation Under Different Bit-Widths

**Experimental Settings.** We compare our approach with state-of-the-art training-oriented quantization methods, including Linear-Quant [9] with different bit-widths, 8bit-Adam (Bnb) [4], 4bit-Adam (Lpmm) [3], and standard FP32 precision. Following the previous work [10], we continually pre-train the TinyLlama-1.1B checkpoint [11] for 400 steps on the PG-19 [12] dataset, chunked into 64k segments, with a context window of 2048. We report the test perplexity in the Proof-pile dataset, and evaluate the trained LLM across four public benchmarks from Huggingface [13], *i.e.,* ARC-Challenge, Hellaswag, Lambada and PIQA. The training settings are provided in Appendix F.

**Results**. As shown in Table 3, we can observe that Linear-Quant [9] exhibits the poorest performance among all compared methods, with its training process being highly unstable and frequently resulting in catastrophic failures. Besides, Bnb [7] exhibits remarkable robustness, achieving performance metrics on par with the full-precision baseline. However, Lpmm Adam [3] shows a notable degradation in performance metrics, even though its training loss is not much different from full precision. Notably, $\pi$-Quant achieves lossless performance in both 13.28-bit and 6.64-bit settings, with metrics even surpassing those of the full-precision baseline. Remarkably, our method reduces state bit-widths

Table 3: Validation of effectiveness under different bandwidths (enclosed in parentheses"()"). The backbone model is TinyLLama. "N/A" denotes the "nan loss" (training collapse). Error bar is provided in Table 6. Results of more LM backbone are reported in Table 8.

| Type | Approach | Memory / Step Time | Train Loss ↓ | Test PPL ↓ | ARC-c ↑ | Hellaswag ↑ | Lambada ↑ | PIQA ↑ | Avg. |
|---|---|---|---|---|---|---|---|---|---|
| Full | FP32 (**32**) | 19.47 GB / 9.77 s | 2.423 | 5.17 | 32.51 | 58.65 | 56.55 | 73.18 | 55.22 |
| High | FP16 (**16**) | 15.53 GB / 8.93 s | $> 10^4$ | 105.36 | 24.57 | 24.52 | 17.19 | 49.02 | 28.83 |
|  | Linear-Quant (**16**) | 15.53 GB / 9.79 s | N/A | N/A | N/A | N/A | N/A | N/A | N/A |
|  | $\pi$-Quant (**13.28**) | 15.40 GB / 11.95 s | **2.420** | 5.15 | 33.02 | **58.78** | **56.76** | 73.23 | **55.45** |
| Medium | Linear-Quant (**8**) | 13.44 GB / 14.97 s | N/A | N/A | N/A | N/A | N/A | N/A | N/A |
|  | Bnb.Adam (**8**) | 13.44 GB / 10.47 s | 2.421 | **5.14** | 33.02 | 58.64 | 56.51 | 73.19 | 55.34 |
|  | $\pi$-Quant (**6.64**) | 13.58 GB / 11.70 s | 2.421 | 5.16 | 33.02 | 58.75 | **56.76** | 73.06 | 55.40 |
| Low | Lpmm.Adam (**4**) | 12.30 GB / 10.66 s | 2.422 | 5.41 | 31.23 | 58.47 | 55.19 | 72.85 | 54.44 |
|  | $\pi$-Quant (**3.32**) | 11.32 GB / 10.56 s | 2.422 | **5.14** | **33.53** | 58.45 | 56.05 | **73.34** | 55.34 |

to 3.32 bits while maintaining similar convergence trend (Figure 3) and performance metrics. These results show that our rotation-based quantization method can effectively reduce the quantization loss, serving as a robust replacement for full-precision optimizers.

Besides, we report the training memory usage and the optimization step time in Table 3. As for training memory, we can observe that Bnb achieves a 30.8% reduction in memory usage compared to the FP32 Adam; Lpmm can further reduce memory usage due to its lower precision representation. In comparison, our method consumes the least memory, achieving a 41.9% reduction in memory usage. As for training speed, quantization-based methods introduce additional quantization and de-quantization operations during training, leading to relatively higher latency. Notably, Bnb and Lpmm Adam have compiled GPU binary search kernels, achieving comparable quantization latency. Since our method requires calculating parameter angles during Quant and Restore stages, resulting in additional computational overhead, which is a limitation of our approach.

## 4.2 Experimental Analysis

### 4.2.1 Accuracy on Other Downstream Tasks

**Experimental Settings.** We evaluate $\pi$-Quant on five tasks on long-range-arena (LRA) benchmark [14], including Listops [15], Text classification on IMDb review dataset [16], Document Retrieval on AAN dataset [17], Pathfinder [18], and Image classification on CIFAR-10 [19]. Besides, we introduce two Seq2Seq tasks, including text summarization on the Samsum [20] and sequential recommendation on the Movielens [21]. Follow the work [22], we use the 2-layer transformer as the backbone, and the training settings are provided in Appendix F. We report the accuracy for the LRA tasks, and report the belu scores and recommendation metrics for the other tasks.

**Results.** The experimental results are shown in Table 4. We can observe that Bnb [4] shows significant accuracy drops on Image, Retrieval, Listops, Summary and RecSys tasks, but surpasses full-precision on Text and PathFinder tasks. Besides, Lpmm [3] shows suboptimal performance across most tasks, with only comparable results on PathFinder. These results suggest that the predefined quantization precision in these methods is not always optimal across different tasks. In comparison, we observe that $\pi$-Quant outperforms the original Adam across all tasks, suggesting that the rotation quantization may represent a superior choice in optimization settings.

### 4.2.2 Ablation Study

We conduct ablation studies to explore the impact of $\bar{\pi}$ settings. As introduced in Section 3.1, the design of $\bar{\pi}$ influences the accuracy of solving for $\theta$ using Eq. (7). To verify this, we change the value of $\bar{\pi}$ as two forms: (1) as a truncated rational number, *i.e.*, $\bar{\pi}_1 = 0.001$ and (2) setting its integer part to other values, *e.g.*, $\bar{\pi}_2 = 3.00100035...$. Specifically, we focus on analyzing quantization errors for two distributions: Gaussian (*i.e.*, $\mathcal{N}(0, 1)$) and uniform distributions (*i.e.*, $\mathbb{U}(0, 1)$). As shown in Figure 4, our method achieves lower error in fitting Gaussian distributions, revealing its non-uniform fitting capability. In addition, $\bar{\pi}$ has a lower error than $\bar{\pi}_1$ as its decimal portion provides error compensation. Besides, we find that setting the integer part of $\bar{\pi}$ to zero yields minimal error.

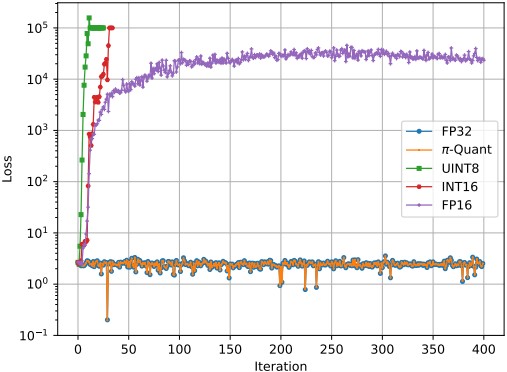

Figure 3: Loss comparison of FP32 and $\pi$-Quant.

Table 4: Performance on other learning tasks. The error bar is presented in Table 7.

| Task | Metric | Adam | Bnb | Lpmm | $\pi$-Quant |
|---|---|---|---|---|---|
| Text | Acc. ↑ | 64.63 | **64.68** | 64.14 | **64.68** |
| | Loss ↓ | 0.634 | **0.630** | 0.635 | 0.631 |
| Image | Acc. ↑ | 40.13 | 39.71 | 39.92 | **41.22** |
| | Loss ↓ | 1.896 | 1.763 | 1.771 | **1.745** |
| Retrieval | Acc. ↑ | 80.70 | 79.92 | 79.19 | **80.79** |
| | Loss ↓ | 43.61 | 45.21 | 45.93 | **44.32** |
| Listops | Acc. ↑ | 38.21 | 37.30 | 37.30 | **38.51** |
| | Loss ↓ | 1.780 | 1.715 | 1.741 | **1.646** |
| PathFinder | Acc. ↑ | 69.15 | 70.83 | **72.05** | 71.92 |
| | Loss ↓ | 55.13 | 53.91 | **49.91** | 58.57 |
| Text Summary | Blue-1 ↑ | **14.42** | 13.34 | 13.71 | 14.32 |
| | Blue-2 ↑ | 8.30 | 7.68 | 7.95 | **8.72** |
| | Blue-3 ↑ | 4.03 | 3.87 | 3.99 | **4.74** |
| | Blue-4 ↑ | 1.54 | 1.61 | 1.57 | **2.02** |
| RecSys | Recall@10 ↑ | 28.19 | 27.75 | 28.01 | **28.76** |
| | Ndcg@10 ↑ | 15.51 | 15.34 | 15.41 | **15.85** |
| | Mrr@10 ↑ | 11.66 | 11.58 | 11.53 | **11.94** |
| | Hit@10 ↑ | 28.19 | 27.75 | 28.01 | **28.76** |

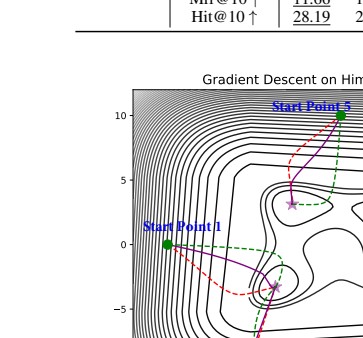

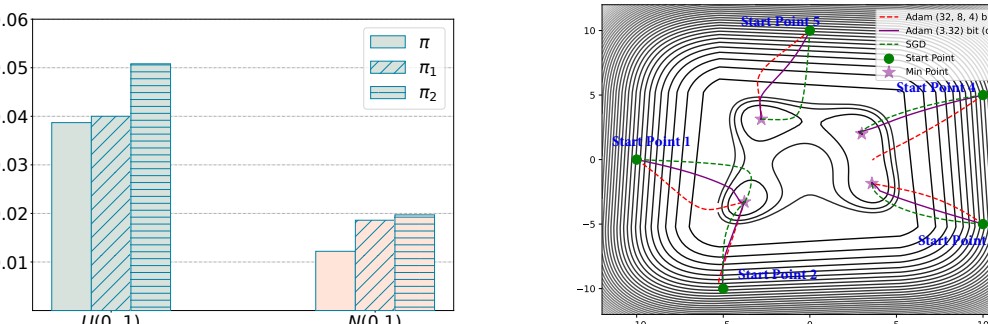

Figure 4: Ablation Study of the $\pi$ setting.

Figure 5: Visualizing the optimization process.

### 4.2.3 Visualizing the Gradient Update Process

To have an intuitive understanding of our approach, we visualize our approach's gradient descent through a simple task: minimizing the Himmelblau's function ($f(x,y) = (x^2 + y - 11)^2 + (x + y^2 - 7)^2$). As shown in Figure 5, we establish multiple starting points and compared the training processes across different optimizers, including Adam with varying bit-widths, SGD, and our proposed method. We can observe that Adam's path is typically shorter than SGD's, showing that Adam's momentum can accelerate convergence. Our method achieves the shortest paths across all initialization points, particularly at start point 4 where it successfully finds the correct minimum while Adam becomes trapped in a local saddle point. These findings show that our rotation method can enhance convergence speed in gradient descent and facilitates escape from local minima and saddle points.

## 5 Related Work

**Model Quantification.** LLMs [1] have achieved outstanding performance across various tasks. However, the extensive parameters in LLMs results in substantial memory usage, significantly increasing the cost of deploying them in real-world scenarios. To this end, a number of quantization methods [23] have been proposed for reducing the deployment cost. Typically, there are two classes of approaches, namely *inference-oriented quantization* and *training-oriented quantization*. Among them, inference-oriented work [24–30] primarily focuses on quantizing the neural weights for achieving faster inference. Through extensive experiments on LLaMA, their approach demonstrates promising results for weight and activation quantization. Despite the progress, these approaches are not suitable to be applied for the training phase, as their precision uniformly across the quantization range, which cannot adapt the non-uniformly distributed parameters. Unlike inference-oriented approaches, training-oriented quantization methods [7, 3] focus on compressing the optimizer states for achieving low training memory. Specifically, they propose to sample the high-frequency elements and then use

a search-based approach to quantize the optimizer states to these values. However, these predefined elements cannot adapt well to all tasks, which may lead to degraded performance in certain tasks.

**Optimization with Complex Rotations.** Traditional neural networks are typically modeled in a real vector space, which exhibits certain constraints when conducting mathematical operations such as exponentiation and vector rotations. To enhance the capability of representation learning, a number of approaches [31–36] have proposed conducting representation learning in complex vector space. Typically, rotation techniques in complex space have been widely applied in the modeling of language models. Specifically, RoPE [34] employ complex rotation to inject the position information into the attention mechanism; xPos [37] employs a two-dimensional pairwise rotation technique to enhance the positional embeddings in Transformers. Unlike these work, we attempt to utilize complex rotations based on irrational numbers for quantization, allowing it to exhibit a non-uniform distribution for parameter optimization.

## 6 Conclusion

In this paper, we proposed a novel optimizer state compression approach $\pi$-Quant. Different from prior work, $\pi$-Quant compressed the parameters using a complex rotation scheme, by leveraging the properties of irrational numbers according to our proposed mathematical findings. In $\pi$-Quant, the two dimensional parameter pairs were efficiently converted into a single rotation angle with our proposed system of geometric equations, which precisely halved the parameter scale with linear complexity. Further, $\pi$-Quant introduced an effective quantization algorithm, which reduced the precision of the rotation angles, all while maintaining full accuracy. In principle, $\pi$-Quant possessed lower quantization error and demonstrated a non-uniform precision distribution, which served as an effective replacement of full precision optimizers. As the further work, we will consider reducing the quantization complexity, and testing the capacity of $\pi$-Quant in more compression scenarios, *e.g.,* KV cache compression in the large language models.

## Acknowledgement

This work was partially supported by National Natural Science Foundation of China under Grant No. 92470205 and 62222215. Xin Zhao is the corresponding author.

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

## A   Proof of Theorem 3.1.

Similar to the original proof, to approximate $z = x + iy$, we construct geometric relationships following Figure 6, thereby deriving the system of geometric equations in Eq. (3):

$$\begin{cases} \alpha = \arctan(y/x), \ \ \beta = \arccos(\sqrt{x^2 + y^2}/2) \\ \theta = \alpha - \beta + 2m\pi, \ m \in \mathbb{Z} \\ \bar{\pi}\theta = \alpha + \beta + 2k\pi, \ k \in \mathbb{Z} \end{cases} \tag{13}$$

Similarly, we can obtain the $\Omega$ to be fitted as:

$$\Omega = \frac{\alpha(1 - \pi) + \beta(1 + \pi)}{2\pi} \tag{14}$$

Within this geometric framework, the proof of the original theorem can be transformed into proving that for given $\epsilon > 0$, there exists an integer $m$ satisfying $|\{m\pi\} - \{\Omega\}| < \epsilon$.

Let $M = 1 - \lfloor \log_{10} \epsilon \rfloor$. We only need to find the integer $m$ such that the first $M$ decimal digits of the fractional parts of $m\pi$ and $\Omega$ are identical. In fact, if we take $\pi$ as a normal irrational number (normal number means that any numerical pattern can be found in the digits of $\pi$).

Remark $T = \lfloor \{\Omega\} \times 10^M \rfloor$ and $S = \mathcal{F}\pi(T)$, where the function $\mathcal{F}\pi : \mathbb{Z}^+ \to \mathbb{N}$ maps a digit sequence to its first occurrence index in $\pi$'s decimal expansion. For example, $\pi = 3.1415926535...$, the mapped values is shown as the following:

| $T$ (Digit Sequence) | $\mathcal{F}_\pi(T)$ | Positions |
|---|---|---|
| 1 | 1 | 3.[1]415... |
| 5 | 4 | 3.141[5]92... |
| 41 | 2 | 3.1[41]592... |
| 159 | 3 | 3.14[159]265... |

In this way, we set $m = 10^{S-1}$, which ensures that the first M decimal digits of the fractional parts of $m\pi$ and $\Omega$ are identical, *i.e.,*

$$|\{m\pi\} - \{\Omega\}| < 10^{-M} < \epsilon/10 < \epsilon. \tag{15}$$

## B   Proof of Lemma 3.2.

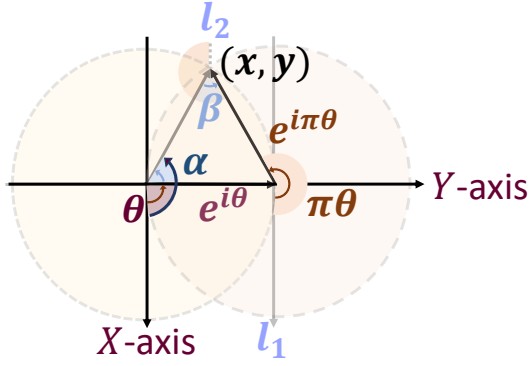

Figure 6: Geometric relationship in $x + iy = e^{i\theta} + e^{i\pi\theta}$

As shown in Figure 6, according to the definition of tangent function, we have:

$$\alpha = \arctan(\frac{y}{x}). \tag{16}$$

Furthermore, since the magnitudes of $e^{i\theta}$ and $e^{i\pi\theta}$ are both 1, according to the cosine theorem, we have:

$$\beta = \arccos(\frac{x^2 + y^2 + 1 - 1}{2 \cdot \sqrt{x^2 + y^2} \cdot 1}) = \arccos(\frac{\sqrt{x^2 + y^2}}{2}). \tag{17}$$

In particular, we also know that the triangle formed by $e^{i\theta}$, $e^{i\pi\theta}$, and $Oxy$ is an isosceles triangle. According to the angular relationship, we can easily deduce that:

$$\alpha - \beta = \theta + 2m\pi, \quad m \in \mathbb{Z}. \tag{18}$$

Additionally, since $l_1$ and $l_2$ are parallel, we have:

$$\alpha + \beta = \pi\theta + 2k\pi, \quad k \in \mathbb{Z}. \tag{19}$$

Therefore, Lemma 3.2 is proved by combing Eq. (16)  Eq. (19).

## C   Proof of Eq. (11).

Recall that $x = \cos\theta + \cos\bar{\pi}\theta$, and we have:

$$
\begin{aligned}
||\Delta x|| &= ||\cos(\theta + \Delta\theta) - \cos\theta + \cos(\bar{\pi}(\theta + \Delta\theta)) - \cos\bar{\pi}\theta|| \\
&= \left|\left|\int_{\theta}^{\theta+\Delta\theta} (\sin\theta + \bar{\pi}\sin\bar{\pi}\theta)d\theta\right|\right| \\
&\leq \int_{\theta}^{\theta+\Delta\theta} \left|\left|(\sin\theta + \bar{\pi}\sin\bar{\pi}\theta)\right|\right| d\theta
\end{aligned}
$$

Let $\theta' = \text{argmax}_\theta \left|\left|(\sin\theta + \bar{\pi}\sin\bar{\pi}\theta)\right|\right|$, we have:

$$
\begin{aligned}
\Delta x &< \Delta\theta \left|\left|\sin\theta' + \bar{\pi}\sin\bar{\pi}\theta'\right|\right| \\
&= \Delta\theta \left|\left|(1 - \bar{\pi})\sin\theta' + \bar{\pi}(\sin\theta' + \sin\bar{\pi}\theta')\right|\right| \\
&= \Delta\theta \left|\left|(1 - \bar{\pi})\sin\theta' + \bar{\pi}y'\right|\right|
\end{aligned} \tag{20}
$$

Since $||y'|| < 1$ (See Eq. (8)), thus we have:

$$\Delta x < \Delta\theta \left|\left|(1 - \bar{\pi}) + \bar{\pi}\right|\right| = \Delta\theta \tag{21}$$

Similarly, we can prove that $\Delta y < \Delta\theta \left|\left|(1 - \bar{\pi})\cos\theta' + \bar{\pi}x'\right|\right| < \Delta\theta$.

## D   Proof of Eq. (12).

Note that the angle space is $[0, 2\pi 10^\lambda]$, remark $M = 2\pi 10^\lambda$, and we have:

$$\mathbb{E}(\Delta x) = \frac{1}{M} \int_0^M 10^{-\lambda} ||\sin\theta + \bar{\pi}\sin\pi\theta||d\theta$$
$$< \frac{10^{-\lambda}}{M} \left( \int_0^M ||\sin\theta||d\theta + \int_0^M \bar{\pi}||\sin\bar{\pi}\theta||d\theta \right)$$
$$= \frac{10^{-\lambda}}{M} \left( \frac{2 \cdot M}{\pi} + \frac{2 \cdot M \cdot \bar{\pi}}{\pi} \right) \tag{22}$$
$$= \frac{2 \cdot (1 + \bar{\pi}) \cdot 10^{-\lambda}}{\pi}$$

Similarly, we have:

$$\mathbb{E}(\Delta y) = \frac{1}{M} \int_0^M 10^{-\lambda} ||\cos\theta + \bar{\pi}\cos\pi\theta||d\theta$$
$$< \frac{10^{-\lambda}}{M} \left( \int_0^M ||\cos\theta||d\theta + \int_0^M \bar{\pi}||\cos\bar{\pi}\theta||d\theta \right)$$
$$= \frac{10^{-\lambda}}{M} \left( \frac{2 \cdot M}{\pi} + \frac{2 \cdot M \cdot \bar{\pi}}{\pi} \right) \tag{23}$$
$$= \frac{2 \cdot (1 + \bar{\pi}) \cdot 10^{-\lambda}}{\pi}$$

## E  Explanation of Eq. (7).

Note that $\bar{\pi}$ can be written as: $\bar{\pi} = 10^{-\lambda} + 10^{-2\lambda} \cdot 0.3589793238$:

$$\bar{\pi} = 0.\underbrace{000\cdots1}_{10^{-\lambda}}\underbrace{0000\cdots}_{10^{-2\lambda}}\underbrace{3589793238\cdots}_{\text{same as } \pi}. \tag{24}$$

Since $m$ retains the first $\lambda$ decimal places of $\Omega$, we have:

$$m \cdot \bar{\pi} = \underbrace{m \cdot 10^{-\lambda}}_{\text{first } \lambda \text{ decimal of } \Omega} + \underbrace{m \cdot 10^{-2\lambda} \cdot 0.3589793238}_{\text{compensation part}}. \tag{25}$$

In this setting, we can guarantee that the difference between $m \cdot \bar{\pi}$ and $\Omega$ does not exceed $10^{-\lambda}$. To further understand the construction of $m$, we present an illustrative example with $\Omega = 1.97525751858\cdots$. If we set $\lambda = 4$, then $m = \lfloor \{\Omega\} \times 10^4 \rfloor = 9752$ (first 4 decimal of $\Omega$). We have:

$$m \cdot \bar{\pi} = 0.\underbrace{9752}_{m \cdot 10^{-\lambda}}\underbrace{350076636569}_{\text{compensation part}} \approx \{\Omega\}. \tag{26}$$

The error is 0.000022510916....

## F  More Experimental Results

Table 5: Hyper-parameter setting of the training pipeline.

| Dataset | PG19 | LRA | Samsum | MovieLens |
|---|---|---|---|---|
| Learning Rate | 2e-5 | 1e-4 | 3e-5 | 1e-3 |
| Learning Rate Schedule | Linear | Linear | Linear | - |
| Weight Decay | 0.0 | 0.0 | 0.0 | 0.0 |
| Batch size | 64 | 256 | 4 | 2048 |
| $\beta_1$ | 0.9 | 0.9 | 0.9 | 0.9 |
| $\beta_2$ | 0.95 | 0.95 | 0.95 | 0.95 |

Table 6: Standard Deviations of LM task

| Metric | ARC-c | Hellaswag | Lambada | PIQA |
|---|---|---|---|---|
| Std (±) | 1.37 | 0.49 | 0.69 | 1.04 |

Table 7: Standard Deviations of Other Machine Learning Tasks

| Task | Text Acc | Image Acc | Retrieval Acc | Listops Acc | PathFinder Acc | BLEU-1 | BLEU-2 |
|---|---|---|---|---|---|---|---|
| Std (±) | 0.0531 | 0.0604 | 0.0410 | 0.1352 | 0.3272 | 0.0640 | 0.0862 |

| Task | BLEU-3 | BLEU-4 | Recall@10 | NDCG@10 | MRR@10 | Hit@10 |
|---|---|---|---|---|---|---|
| Std (±) | 0.0780 | 0.0462 | 0.3585 | 0.2927 | 0.1716 | 0.3585 |

Table 8: Model Comparison of Other Backbone LMs

| Model | Method | Memory (GB) | Step Time (s) | ARC-c | Hellaswag | Lambada | PIQA |
|---|---|---|---|---|---|---|---|
| GPT-2 | FP32 | 6.34 | 5.22 | 22.95 | 31.14 | 32.56 | 62.51 |
| | Bnb.Adam (8-bit) | 4.58 | 5.86 | 23.05 | 31.14 | 32.06 | 62.37 |
| | Lpmm.Adam (4-bit) | 3.39 | 5.37 | 21.67 | 30.82 | 33.02 | 60.14 |
| | $\pi$-Quant (3.32-bit) | 3.07 | 6.14 | 22.83 | 31.14 | 32.75 | 62.23 |
| RoBERTa | FP32 | 4.93 | 1.61 | – | 23.49 | – | 50.27 |
| | Bnb.Adam (8-bit) | 4.22 | 1.67 | – | 23.40 | – | 50.30 |
| | Lpmm.Adam (4-bit) | 3.79 | 1.64 | – | 22.73 | – | 48.53 |
| | $\pi$-Quant (3.32-bit) | 3.57 | 1.83 | – | 23.37 | – | 50.27 |
| LLaMA-7B | FP32 | 55.11 | 19.27 | 50.51 | 76.21 | 73.55 | 79.16 |
| | Bnb.Adam (8-bit) | 42.71 | 19.44 | 50.55 | 75.95 | 73.55 | 78.96 |
| | Lpmm.Adam (4-bit) | 36.42 | 19.35 | 48.63 | 74.99 | 73.02 | 77.36 |
| | $\pi$-Quant (3.32-bit) | 32.87 | 20.15 | 50.32 | 76.48 | 73.40 | 79.07 |

*Note*: Full reproducibility details are in the supplementary code, '–' denotes the `NaN` encountered in the output.

