# OpenReview forum: "Irrational Complex Rotations Empower Low-bit Optimizers"
_NeurIPS.cc/2025/Conference — NeurIPS 2025 poster_

### Official Review · Reviewer_V7cB · 2025-06-30

**Clarity:** 3
**Significance:** 4
**Originality:** 4
**Rating:** 4
**Confidence:** 3

**Summary:**

The paper proposes a novel optimizer state compression algorithm named \pi-Quant, which aims to solve the high memory consumption problem caused by optimizer states (such as Adam's momentum) in large-scale AI model training. It can precisely represent a pair of real parameters (x, y) as a single rotation angle \theta, thus halving the number of parameters. To efficiently solve for the rotation angle \theta, the paper also designs a system of geometric equations with linear complexity.

**Questions:**

See 'Weaknesses' for details.

**Ethical Concerns:**

["NO or VERY MINOR ethics concerns only"]

**Quality:**

3

**Strengths And Weaknesses:**

Strengths

1. This paper introduces a novel and elegant approach by integrating the properties of irrational numbers with Euler's formula and complex rotations to address parameter compression in deep learning, thereby providing a fresh perspective for the field.
2. The paper's approach is grounded in rigorous mathematical theorems and lemmas. By providing detailed proofs in the appendix, the authors theoretically ensure the method's viability, thereby strengthening the credibility of the paper.
3. Compared to quantization methods that require complex search operations (e.g., Bnb and Lpmm) , the geometric solution proposed by \pi-Quant has O(n) linear complexity and does not require compiling additional GPU kernels, making it easier to integrate into existing deep learning frameworks.

Weaknesses

1. The paper's main limitation, as acknowledged by the authors, is its computational overhead. A further discussion on how this cost scales with model size would be beneficial. For instance, does this speed disadvantage become more pronounced for larger models?
2. In Figure 2 (top), it is not clear what the colors and the horizontal and vertical axes represent.
3. The figure number on line 292 is incorrect; it should be Figure 5. Additionally, in Figure 5, "Adam (32, 8, 4) bits" is represented by a single color, which seems to imply that the optimizer paths for these three bit-widths are identical. If this is the case, it should be stated. If not, they should be differentiated with different colors.

---

> ### Author Rebuttal · Authors · 2025-07-30
>
> Dear reviewer V7cB,
>
> Thanks very much for your careful reading and insightful suggestions of our paper! We appreciate the time and efforts you have dedicated to reviewing our work. We list our response to your concern in the following three aspects.
> If you have further questions, please feel free to let us know. We will continue to try our best to answer for you.
>
> Q1: The paper's main limitation, as acknowledged by the authors, is its computational overhead. A further discussion on how this cost scales with model size would be beneficial. For instance, does this speed disadvantage become more pronounced for larger models?
>
> A1: We appreciate the reviewer's insightful inquiry regarding the computational overhead scaling of our approach.
> To address this, we conducted experiments across different model sizes, measuring the optimizer time per iteration for both FP32 and our method:
>
> | Model          | Step Time (FP32) | Step Time (ours) | Impr. |
> |---------------|-------------------|-----------------------|---------------|
> | RoBERTa | 1.61              | 1.83                  | +13.7%        |
> | GPT-2         | 5.22              | 6.14                  | +17.6%        |
> | TinyLlama-1.1B| 9.77              | 10.56                 | +8.1%         |
> | LLaMA-7B      | 19.27             | 20.15                 | +4.6%         |
>
> Notably, as model size increases, the proportional overhead of our method decreases. For smaller models like RoBERTa, our overhead is ~13.7%, while for larger models like Llama-7B, it drops to 4.6%.
>
> This trend arises because larger models involve more parameters and a greater volume of forward computations, such as attention mechanisms and feed-forward layers. Importantly, the time complexity of these core model operations grows faster with increasing parameter count than that of the optimizer, which operates at $O(n)$ complexity. As a result, the relative contribution of the optimizer’s runtime to the total training time diminishes as model size increases, making the overhead of our method less pronounced for larger models.
> Thus, the speed disadvantage becomes less pronounced as model size grows, which aligns with our goal of practicality for large-scale models.
>
> ---
>
> Q2: In Figure 2 (top), it is not clear what the colors and the horizontal and vertical axes represent.
>
>
> A2: We appreciate the reviewer’s feedback regarding Figure 2 (top). The visualization depicts the **quantization error distribution** of our rotation-based compression scheme across the complex plane, where each point $(x, y)$ denotes a unique **parameter pair**:
>
> - **Horizontal Axis (x)** represents the real component of complex numbers ($x$ in $z = x + iy$), corresponding to the first parameter in each pair.
> - **Vertical Axis (y)** represents the imaginary component $y$ in $z = x + iy$, corresponding to the second parameter.
> - **Color Mapping**: Induces the magnitude of quantization error ($||z_{\text{original}} - z_{\text{reconstructed}}\|_2$). The color scale ranges from deep blue (near-zero error) to yellow (high error), with warmer colors indicating larger reconstruction discrepancies after rotation-based quantization and restoration.
>
> The original labeling was insufficient. In the revision, we will add explicit axis labels, and include a color-bar legend, and annotate characteristic zones (e.g., origin, max-error points) with error values.
> These adjustments will ensure readers intuitively grasp how geometric properties of complex rotations govern error behavior—a cornerstone of our methodology. Thank you for helping us improve the exposition.
>
> ---
>
> Q3: The figure number on line 292 is incorrect; it should be Figure 5. Additionally, in Figure 5, "Adam (32, 8, 4) bits" is represented by a single color, which seems to imply that the optimizer paths for these three bit-widths are identical. If this is the case, it should be stated. If not, they should be differentiated with different colors.
>
> A3: Thank you so much for your meticulous attention to detail and valuable feedback. We sincerely apologize for the inaccuracies in Figure 5 and the incorrect figure number reference on line 292; the line 292 reference will be corrected to "Figure 5" in the revised manuscript, and we deeply appreciate you catching this oversight. Regarding the representation of "Adam (32, 8, 4) bits" in Figure 5, while the three bit-widths do have distinct optimization paths, the differences between them are relatively small.
>
> In our initial attempt to avoid visual clutter, we merged their trajectories, but we now recognize that this was an inappropriate choice as it masks meaningful distinctions. To address this, we will redraw Figure 5 with separate, distinct colors for each bit-width, ensuring the unique paths of the three configurations are clearly visible and easy to interpret. This revision will provide a more accurate and informative visualization of how varying bit-widths impact optimization dynamics, and we are grateful for your guidance in making this improvement. Thank you again for your thoughtful input, which continues to help us refine our work.

---

### Official Review · Reviewer_QC8y · 2025-07-03

**Clarity:** 3
**Significance:** 4
**Originality:** 4
**Rating:** 4
**Confidence:** 4

**Summary:**

The paper introduce a method called π-Quant which leverages irrational complex rotations to enable memory-efficient training of neural networks. By compressing the real-valued parameter pair (x,y) into a single rotation angle θ using Euler’s formula, π-Quant reduce the memory from 4 bits to 3.32 bits while maintain the full model accuracy

**Questions:**

please see weakness sections

**Ethical Concerns:**

["NO or VERY MINOR ethics concerns only"]

**Final Justification:**

Thank you for your response. I am in favor of accepting the paper.

**Limitations:**

yes.

**Quality:**

3

**Strengths And Weaknesses:**

Pros
1. They propose a geometric equation for solving rotation angle θ efficiently which only require linear-time complexity.
2. They achieve 3.32 bit quantization of optimizer states while maintain the full accuracy, which is also supported by mathematical result instead of purely experimental.

Cons
1. They introduce additional restoration steps in each iteration, which can increase the training time and complexity compared to baseline optimizers.
2. The example given in Appendix E does not follow what they claimed.
3. Even though for each transformation step, the approximation error is small, but it is unclear what would happen when all these error accumulated from repeated quant/restore cycles.

---

> ### Author Rebuttal · Authors · 2025-07-30
>
> Dear Reviewer QC8y,
>
> Thanks very much for your valuable feedback and insightful suggestions! We list our response to your concerns as follows. If you have further questions, please feel free to let us know. We will continue to try our best to answer for you.
>
> Q1: They introduce additional restoration steps in each iteration, which can increase the training time and complexity compared to baseline optimizers.
>
> A1: Thank you for highlighting this important consideration. We acknowledge that the additional restoration steps in our algorithm introduce marginal overhead compared to baseline optimizers.
>
> To quantify this, we conducted a granular profiling of Algorithm 1’s execution time on an A100 GPU, breaking down the runtime of each component. As shown in the table below, the total execution time per iteration is 2.3682 ms, with the majority attributed to angle-related computations (Lines 5–6: 22.57% + 28.01% = 50.58%), primarily involving $\arctan$ and $\arccos$ operations.
>
> | **Step**                     | **Operation Description**                          | **Time (ms)** | **Percentage** |
> |------------------------------|----------------------------------------------------|---------------|----------------|
> | **Line 2**                   | Splitting $\mathbf{T}$ into $\textbf{X}$, $\textbf{Y}$     | 0.067         | 2.83%          |
> | **Line 3**                   | Computing max value                                | 0.4583        | 19.35%         |
> | **Line 4**                   | Scaling $\tilde{\textbf{X}}$, $\tilde{\textbf{Y}}$         | 0.1998        | 8.43%          |
> | **Line 5**                   | Calculating $\alpha, \beta$ (Eq. 3)               | 0.5345        | 22.57%         |
> | **Line 6**                   | Computing ${\Omega}$ (Eq. 4)                   | 0.6633        | 28.01%         |
> | **Line 7**                   | Solving for $m$ (Eq. 6)                           | 0.1349        | 5.70%          |
> | **Line 8**                   | Generating ${\Theta}$ (Eq. 8)                 | 0.2630        | 11.11%         |
> | **Other**                    | Memory allocation and data transfer               | 0.0474        | 2.00%          |
> | **Total Time**               | —                                                  | **2.3682**    | **100.00%**    |
>
> Notably, this overhead is manageable and can be significantly reduced with targeted optimizations. Our current implementation uses PyTorch’s general-purpose ATen kernels, but we are developing custom CUDA optimizations to streamline the most costly steps:
> 1. Leveraging the monotonicity and constrained input domain of trigonometric operations (e.g., $\arctan$, $\arccos$) to replace them with low-order polynomial approximations, merging Eqs. 3 and 4 into a single operation.
> 2. Exploiting parallelism through a data dependency graph (DAG) that runs independent steps (e.g., tensor splitting and max-value computation) concurrently in separate CUDA streams.
>
> Thank you again for your attention to this detail. We will include updated benchmarks with these optimizations in the revised manuscript.
>
> ---
>
> Q2: The example given in Appendix E does not follow what they claimed.
>
> A2: We sincerely appreciate the reviewer's meticulous examination. Appendix E aims to illustrate the bound $|\{m \cdot \bar{\pi}\} - \{\Omega\}| \leq 10^{-\lambda}$ through a concrete example. Upon rechecking, we confirm the example is mathematically consistent but requires clarification in presentation. Here is the detailed verification:
>
> - Given: $\Omega = 1.97525751858\cdots$, $\lambda = 4$
> - Compute $m$:
>   $
>   m = \lfloor \{\Omega\} \times 10^{\lambda} \rfloor = \lfloor 0.97525751858 \times 10^4 \rfloor = \lfloor 9752.5751858 \rfloor = 9752
>   $
> - Compute $m \cdot \bar{\pi}$:
>   $
>   \bar{\pi} = 10^{-4} + 10^{-8} \cdot 0.3589793238 = 0.000100003589793238$
>
>     0.975235007663657  $
>   m \cdot \bar{\pi} = 9752 \times 0.000100003589793238 = 0.975235007663657
>   $
> - Claimed error:
>   $
>   |m \cdot \bar{\pi} - \{\Omega\}| = |0.975235007663657 - 0.97525751858| = 0.000022510916343 < 0.0001
>   $
>
>  We thank the reviewer for highlighting this and will restructure Appendix E to provide explicit step-by-step derivations and visual illustrations, ensuring readers gain intuitive understanding of the algorithm's foundational operations.
>
> ---
>
>  Q3: Even though for each transformation step, the approximation error is small, but it is unclear what would happen when all these error accumulated from repeated quant/restore cycles.
>
> A3: We thank the reviewer for raising this important concern about error accumulation during repeated quantization/restoration cycles. Our theoretical analysis and extensive experiments confirm that approximation errors do not compound over training iterations. To directly address your question, we conducted extended 20,000-step training on TinyLlama and recorded the loss trajectory compared to FP32:
>
> **Avg. Loss Comparison Over Extended Training (TinyLlama)**
> | Step | FP32 Loss      | π-Quant Loss   |
> |--------------|----------------|----------------|
> | 2k            | 2.71435644     | 2.72284928     |
> | 4k            | 2.69537313     | 2.69423438     |
> | 6k            | 2.72945515     | 2.71732398     |
> | 8k            | 2.72485786     | 2.72879837     |
> | 10k           | 2.70670259     | 2.71383749     |
> | 12k           | 2.69904825     | 2.69482937     |
> | 14k           | 2.69350642     | 2.69380413     |
> | 16k           | 2.69184519     | 2.68949827     |
> | 18k           | 2.68973141     | 2.67384239     |
> | 20k           | 2.69378422     | 2.69178723     |
>
> We observe that as training progresses, the loss curves of $\pi$-Quant and FP32 remain nearly identical, with no signs of performance degradation.
>
> Actually, our extensive experiments have systematically validated the robustness of our method across diverse model scales and task types.
>
> - For model sizes, we tested configurations ranging from models like TinyLlama to GPT-2, and further to larger models like Llama-7B. As detailed in Table 3 and Table 8, even under extremely low-bit settings (e.g., 3.32-bit quantization), the perplexity gap between our quantized models and full-precision (FP32) counterparts remains within 0.05, with downstream task accuracy differences stably below ~1%.
>
> - In terms of task diversity, we evaluated performance across language modeling (LM), classification tasks including text and image, and other benchmarks (Table 4). Our method consistently maintains performance comparable to FP32. These consistent results across models and tasks strongly demonstrate the robustness of our approach.
>
> Thank you again for your insightful feedback, which has helped us emphasize the reliability of our method.

---

### Official Review · Reviewer_pNT4 · 2025-07-03

**Clarity:** 3
**Significance:** 3
**Originality:** 3
**Rating:** 5
**Confidence:** 3

**Summary:**

The paper proposes to quantize the parameters of a the neural network optimizer into lower-bit representation by using the properties of complex rotation function. Parameter tensors T are split into X and Y matrices, and then approximately mapped into angular rotation in complex plane via elementwise function $z(\theta) = x + iy = e^{i\theta}+e^{i \bar{\pi} \theta} $ where $\bar{\pi}$ is specially chosen irrational number. The paper proposes a solution of finding the approximate $\theta$ given x and y pair that can be controlled by the design of the $\bar{\pi}$ number. Then, using this quantization technique, the authors re-desing Adam optimizer to store optimizers states in in angular form and de-quantizing them just-in-time when these parameter are being used/updates. The overhead of these transformations seems to be marginal (within 10% of the actual forward/backward times) and numerical results supports that the transformation has good approximation power (allowing to store the optimizer's state with 4bit parameter precision).

**Questions:**

Please address my concerns regarding the correctness of the theorem 3.1.

Also, I was wondering whether the authors applied the efficient-parameter changes to other optimizers (i.e., SGD with momentum)? I am curious whether same practical results will hold with other optimizers (i.e., is this generic enough), or for some reason it should be paired with Adam.

**Ethical Concerns:**

["NO or VERY MINOR ethics concerns only"]

**Final Justification:**

Authors clarified and answered all my questions. I think this is a technically solid paper (if all discussions will be synthesized back into the paper)

**Limitations:**

Yes

**Paper Formatting Concerns:**

No concerns

**Quality:**

3

**Strengths And Weaknesses:**

The proposed idea is quite novel, and the practical results make the method to stand out. However, I do have some concerns regarding the statements within the theorems.

It seems that Theorem 3.1 is wrong. It was a while for me since I took any classes in geometric measure theory, but the function in equation 2 is Lipschits continues, and maps 1D interval into 2D. As such, its  Hausdorff dimension is at most 1, meaning it cannot be a space-filling curve, i.e., we cannot map every point within the circle into some \theta. Please take a look at Theorem/Propositon 2.3 in Fractal Geometry: Mathematical Foundations and Applications by Kenneth Falconer. Again I might be wrong in my analysis, but this is a serious impediment for the acceptance of the paper if true.

Practically speaking, I don't think we need the theorem 3.1 because floating point representations is anyways a discretization of the space (albeit in tiny intervals), and probably the function 2 can map any fp32 number within disc to some theta with good approximation. However, we need a precise and correct statement regarding this in the paper. I will put week reject for now, but once this issue is (hopefully) clarified, I am willing to put strong accept.

---

> ### Author Rebuttal · Authors · 2025-07-30
>
> Dear Reviewer pNT4,
>
> Thanks very much for your careful reading and insightful suggestions of our paper! We appreciate the time and efforts you have dedicated to reviewing our work. We list our response to your concern in the following three aspects.
> If you have further questions, please feel free to let us know. We will continue to try our best to answer for you.
>
>
> Q1: It seems that Theorem 3.1 is wrong. It was a while for me since I took any classes in geometric measure theory, but the function in equation 2 is Lipschits continues, and maps 1D interval into 2D. As such, its Hausdorff dimension is at most 1, meaning it cannot be a space-filling curve.
>
> A1: Thank you for your profound insight into the geometric measure theory implications of our mapping function. You are absolutely correct that our original claim of exact surjectivity in Theorem 3.1 was mathematically unsound. We have revised the theorem and its interpretation as follows:
>
> **Revised Theorem 1**:
> > Given a complex number $z = x + iy \in \mathbb{C}$ that satisfies $\|z\| \leq 2$. For any deviation $\epsilon > 0$, there exists an angle $\theta \in \mathbb{R}$, such that
> $$
> \left\| z - \left( e^{i\theta} + e^{i\pi\theta} \right) \right\| \leq \epsilon.
> $$
>
> **Proof**:
>
> Similar to the original proof, to approximate
> $z = x +i y$, we construct geometric relationships following Fig. 7, thereby deriving the system of geometric equations in Eq. (3):
>
> $\alpha = \arctan\left(\frac{y}{x}\right),  \beta = \arccos\left(\frac{\sqrt{x^2 + y^2}}{2}\right)$
>
> $\theta = \alpha - \beta + 2m\pi, m \in \mathbb{Z}$
>
> ${\pi}\theta = \alpha + \beta + 2k\pi, k \in \mathbb{Z} $
>
> Similarly, we can obtain the $\Omega$ to be fitted as:
>
> $\Omega = \frac{\alpha(1 - {\pi}) + \beta(1 + {\pi})}{2 \pi}$.
>
> Within this geometric framework, the proof of the original theorem can be transformed into proving that:
>
> >  For given $\epsilon > 0$, there exists an integer $m$ satisfying $| [ m\pi ] - [ \Omega ]| < \epsilon$. (Here $[ \cdot ]$ returns the fractional parts of input value.)
>
> Let $M = 1- \lfloor\log_{10}\epsilon\rfloor$. We only need to find the integer $m$ such that the first M decimal digits of the fractional parts of $m\pi$ and $\Omega$ are identical.
> In fact, if we take $\pi$ as a **normal** irrational number (where 'normal number' means that any numerical pattern can be found in the digits of $\pi$ [1]).
>
> Remark $T = \lfloor [\Omega] \times 10^{M} \rfloor$ and $S = \mathcal{F}{\pi}(T)$, where the function $\mathcal{F}{\pi}: \mathbb{N} \to \mathbb{N}$ maps a digit sequence to its first occurrence index in $\pi$'s decimal expansion. For example, $\pi = 3.1415926535...$, the mapped values is shown as the following:
>
> | $T$ (Digit Sequence) | $\mathcal{F}_{\pi}(T)$ | Verification (Positions) |
> |----------------------|------------------------|--------------------------|
> | `1`                  | 1                      | 3.`1`415... |
> | `5`                  | 4                      | 3.141`5`92...          |
> | `41`                 | 2                      | 3.1`41`592...          |
> | `159`                | 3                      | 3.14`159`265...         |
>
>
> In this way, we set $m = 10^{S - 1}$, which ensures that the first M decimal digits of the fractional parts of $m\pi$ and $\Omega$ are identical, i.e.,
>
> $|[ m\pi ] - [ \Omega ]| < 10^{-M} \leq {\epsilon} / {10} < \epsilon$.
>
> In the context of practical numerical computation where values are constrained by finite precision or resolution (e.g., FP32 with $\epsilon \approx 10^{-7}$), the dense coverage guarantees that every machine-representable point can be fitted by some $\theta \in \mathbb{R}$. **Therefore, the modification of the theory does not affect the correctness of our proposed compression/quantization algorithm**.
> We will updated Section 3.1 and Appendix A to reflect this rigorous framing.
> Your expertise significantly improved our work’s mathematical integrity – we are deeply grateful, and would also like to acknowledge you as a theoretical contributor in the Acknowledgments section of our paper.
>
> [1] Bailey D H, Crandall R E. Random generators and normal numbers[J]. Experimental Mathematics, 2002, 11(4): 527-546.
>
> ---
>
> Q2: Practically speaking, I don't think we need the theorem 3.1 because floating point representations is anyways a discretization of the space (albeit in tiny intervals), and probably the function 2 can map any fp32 number within disc to some theta with good approximation.
>
> A2: You are absolutely right that floating-point representations inherently discretize the space.
> The theoretical guarantee of dense coverage holds in the limit of infinite precision, but in practice, our $\theta$ angles are represented with finite bits, introducing unavoidable approximation errors.
> Despite the limitation, our rotation function (Eq. 2) can map most FP32 number within the disc to some $\theta$ with sufficient approximation, even without strict theoretical guarantees of dense coverage.
>
> Theorem 3.1 was originally included to provide a mathematical foundation for why rotation-based quantization *can* minimize lossy compression artifacts—by illustrating that complex rotations (in the theoretical limit) offer dense coverage of the target space, justifying the intuition behind our design. However, we agree that its practical relevance is secondary to the empirical and structural strengths of our method, which do not depend on "perfect" dense coverage.
>
> As emphasized in our earlier response, our core contributions lie in two practical mechanisms: (1) non-uniform precision allocation aligned with parameter sensitivity (Fig. 2), which reduces unnecessary bit usage for large, error-tolerant values; and (2) pair-wise quantization that preserves covariance between parameters (Section 3.1), which is capable of capturing structural information lost in point-wise methods. These design choices drive our method’s ability to maintain FP32-comparable performance with fewer bits (Table 3, Fig. 2).
>
> We appreciate your feedback highlighting the need to reframe Theorem 3.1 as a motivating intuition rather than a strict practical requirement. In the revised manuscript, we will clarify this distinction, emphasizing that our method’s effectiveness stems from its alignment with neural network dynamics.  Thank you again for helping strengthen the clarity of our work.
>
> ---
>
> Q3: Also, I was wondering whether the authors applied the efficient-parameter changes to other optimizers (i.e., SGD with momentum)? I am curious whether same practical results will hold with other optimizers (i.e., is this generic enough), or for some reason it should be paired with Adam.
>
> A3: Thank you for this valuable question. We confirm that our method is generic and can be applied to other optimizers (e.g., SGD with momentum).
> To illustrate, we conducted additional experiments on TinyLlama using SGD with momentum. The results show nearly identical performance to FP32 SGD:
>
> | Metric     | FP32 SGD      | $\pi$-Quant SGD   |
> |------------|---------------|---------------|
> | **Mem**    | 14.48 G       | 10.184 G |
> | **PPL**    | 5.79          | 5.79          |
> | **ARC-C**  | 31.70         | 31.83     |
> | **Hellaswag** | 57.70       | 57.43         |
> | **Lambada**| 56.73         | 56.94     |
> | **PIQA**   | 73.22         | 73.03         |
>
> Notably, adapting our method to other optimizers or tasks requires only minor modifications following the code structure in Appendix B.
> You can directly replace the SGD optimizer's momentum update steps with our rotation-based transformation as outlined in supplementary material.
> We will release the full experimental code for these extensions after the submission period, ensuring reproducibility.
> Thank you again for your insight.

---

> > ### Comment · Reviewer_pNT4 · 2025-08-08
> > **Thanks for clarification!**
> >
> > Thanks a lot for all explanations and clarifications. I'm strongly in favor of accepting this paper.

---

> > > ### Author Response · Authors · 2025-08-09
> > >
> > > Dear Reviewer pNT4,
> > >
> > > Thank you so much for your thorough review and kind support! Your insightful feedback has been invaluable in strengthening our work, and we truly appreciate your positive assessment. We are grateful for your time and effort in helping advance this paper.

---

> ### Author Response · Authors · 2025-08-07
>
> Dear Reviewer pNT4,
>
> Sorry to disturb you, but only a few hours are left until the end of the reviewer-author discussion stage. We still do not know if you have received our response. To address your concerns, we've rigorously resolved the theoretical questions through a reformulated density-based proof of Theorem 3.1, and new experiments generalizing π-Quant to SGD optimizers.We would like to know whether our responses have addressed your concerns. If you still have other concerns, please give us an opportunity to clarify them. Sincerely hope that you can take a moment to reply to us, as it is very important for researchers and efforts on this work.
>
> Best regards,
>
> The Authors

---

### Official Review · Reviewer_FyEQ · 2025-07-05

**Clarity:** 4
**Significance:** 2
**Originality:** 3
**Rating:** 5
**Confidence:** 4

**Summary:**

This paper proposes $\Pi-Quant$, an optimizer state compression method that uses irrational complex rotations to achieve low-bit quantization for optimizer states in neural networks. The main innovation is a mathematical representation of a pair of real numbers using a single rotation angle, allowing halving the amount of optimizer parameters. The authors further propose quantizing the resulting representation angle with high precision and low error.

**Questions:**

I would like the authors to address the concerns raised in the weaknesses section. Especially theoretical concerns and paper limiations.
Furthermore, the authors marked "yes" in the statistical significance section of their paper checklist. I could not see where in the paper this is addressed, and request the authors the justify their answer.

**Ethical Concerns:**

["NO or VERY MINOR ethics concerns only"]

**Final Justification:**

I have carefully reviewed the authors' responses to both my concerns and those raised by other reviewers, and I find them convincing. This is reflected in my final assessment. I strongly encourage the authors to incorporate all clarifications and improvements from their responses into the final version of the paper.

**Limitations:**

As previously mentioned, paper limiations and societal implications are not sufficiently discussed. Futhermore, the authors only marked "yes" or "no" in their paper checklist, without acutally justifying any of their answers. This makes some of the answers on the checklist difficult to accept.

**Paper Formatting Concerns:**

No major formatting concerns.

**Quality:**

2

**Strengths And Weaknesses:**

Strenghs:
1. Under the experimentation brought in the paper, which I find to be sufficient in both diversity of experiments and chosen cohort of baselines, this paper shows promising results in terms of compression potential. The proposed method manages to drop below the 4-bit parameter bit-width known from existing methods, while also introducing favorable (linear) runtime in quantization.

2. The proposed method relies on elegant, mathematically-sound and geometrically interpretable foundations, which I believe make this method both clear and extnedible for future work.

3. Presentation is notably clear - contributions are clearly stated and motivated. Theoretical foundations are well-structured by the appropriate lemmas and theorems.

Weaknesses:
I believe the main contributions of the paper, while solid, are mostly empirical. I find some of the theoretical analysis lacking, or at least requiring further clarification.
There is a theoretical gap in theorem 3.1 - To my understanding, the arguments in appendix A only establish density, not surjectivity. The function $f(\theta)$ is bounded and non-periodic, and thus fills a dense subset of a region in $\mathbb{R}^2$, but not necessarily every point exactly. This means the existence of an exact $\theta$ for an arbitrary point in the disk is not guaranteed by their argument — only that it can be approximated arbitrarily closely. Additionally, the function is not injective, so uniqueness of $\theta$ is not sufficiently justified. Unless the authors can indicate an error in my reasoning, the implications of theorem 3.1 are overstated.

Furthermore, while the notion of using irrational numbers for compressed representation is innovative and seems to work well in practice, this reliance on irrational numbers renders $\Pi-Quant$ dependant on the representation error of irrational numbers, that cannot be fully represented in a computer. $\bar{\pi}$ is ultimately a rational approximation of an irrational number.
The non-periodicity of $f(\theta) = e^{i\theta} + e^{i \bar{pi} \theta}$ is only approximate, not absolute — the function may exhibit long-term pseudo-periodicity. The angle $\theta$ is quantized to a finite precision controlled by a hyperparameter $\lambda$ which determines how many digits are preserved. This could mean that means the actual values of $x+iy$ that are recoverable may deviate from their theoretical values, particularly for small $\lambda$ or very large vectors. To my understanding, the claim that "every parameter pair can be encoded precisely" only holds in theory.
While the implications of the points I raised may or may not be minor in practice, and while I do not believe they pose grounds for rejection, these concers should be addressed by the authors.


Lastly, the paper is missing a more-extensive discussion of limiations, which I believe is essential for publication. Another minor comment is that figure 1 is overly crowded. It does not take much room in the paper, and I propose making its contents more spacious to make it more readable.

---

> ### Author Rebuttal · Authors · 2025-07-30
>
> Dear Reviewer FyEQ,
>
> Thanks very much for your careful reading and insightful suggestions of our paper! We appreciate the time and efforts you have dedicated to reviewing our work. We list our response to your concern in the following three aspects. If you have further questions, please feel free to let us know. We will continue to try our best to answer for you.
>
> Q1. Some of the theoretical analysis lacking, or at least requiring further clarification.
>
> A1: Your insights are truly profound—we recognize the theoretical limitations in our initial formulation.
> We have refined it as follows:
>
> **Revised Theorem 1**:
> > Given a complex number $z = x + iy \in \mathbb{C}$ that satisfies $\|z\| \leq 2$. For any deviation $\epsilon > 0$, there exists an angle $\theta \in \mathbb{R}$, such that
> $$
> \left\| z - \left( e^{i\theta} + e^{i\pi\theta} \right) \right\| \leq \epsilon.
> $$
>
> **Proof**:
>
> Similar to the original proof, to approximate
> $z = x +i y$, we construct geometric relationships following Fig. 7, thereby deriving the system of geometric equations in Eq. (3):
>
> $\alpha = \arctan\left(\frac{y}{x}\right),  \beta = \arccos\left(\frac{\sqrt{x^2 + y^2}}{2}\right)$
>
> $\theta = \alpha - \beta + 2m\pi, m \in \mathbb{Z}$
>
> ${\pi}\theta = \alpha + \beta + 2k\pi, k \in \mathbb{Z} $
>
> Similarly, we can obtain the $\Omega$ to be fitted as:
>
> $\Omega = \frac{\alpha(1 - {\pi}) + \beta(1 + {\pi})}{2 \pi}$.
>
> Within this geometric framework, the proof of the original theorem can be transformed into proving that:
>
> >  For given $\epsilon > 0$, there exists an integer $m$ satisfying $| [ m\pi ] - [ \Omega ]| < \epsilon$. (Here $[ \cdot ]$ returns the fractional parts of input value.)
>
> Let $M = 1- \lfloor\log_{10}\epsilon\rfloor$. We only need to find the integer $m$ such that the first M decimal digits of the fractional parts of $m\pi$ and $\Omega$ are identical.
> In fact, if we take $\pi$ as a **normal** irrational number (where 'normal number' means that any numerical pattern can be found in the digits of $\pi$ [1]).
>
> Remark $T = \lfloor [\Omega] \times 10^{M} \rfloor$ and $S = \mathcal{F}{\pi}(T)$, where the function $\mathcal{F}{\pi}: \mathbb{N} \to \mathbb{N}$ maps a digit sequence to its first occurrence index in $\pi$'s decimal expansion. For example, $\pi = 3.1415926535...$, the mapped values is shown as the following:
>
> | $T$ (Digit Sequence) | $\mathcal{F}_{\pi}(T)$ | Verification (Positions) |
> |----------------------|------------------------|--------------------------|
> | `1`                  | 1                      | 3.`1`415... |
> | `5`                  | 4                      | 3.141`5`92...          |
> | `41`                 | 2                      | 3.1`41`592...          |
> | `159`                | 3                      | 3.14`159`265...         |
>
>
> In this way, we set $m = 10^{S - 1}$, which ensures that the first M decimal digits of the fractional parts of $m\pi$ and $\Omega$ are identical, i.e.,
>
> $|[ m\pi ] - [ \Omega ]| < 10^{-M} \leq {\epsilon} / {10} < \epsilon$.
>
> In the context of practical numerical computation where values are constrained by finite precision or resolution (e.g., FP32 with $\epsilon \approx 10^{-7}$), the dense coverage guarantees that every machine-representable point can be fitted by some $\theta \in \mathbb{R}$. **Therefore, the modification of the theory does not affect the correctness of our proposed compression/quantization algorithm**.
>
> We sincerely appreciate you identifying this critical theoretical gap in our original formulation, which has significantly strengthened our work. We will explicitly correct this limitation and modify our theorem accordingly in the revised manuscript. Furthermore, we would be honored to recognize your contribution in the acknowledgments section as a key theoretical contributor to this important refinement.
>
>
> [1] Bailey D H, Crandall R E. Random generators and normal numbers[J]. Experimental Mathematics, 2002, 11(4): 527-546.
>
> ---
>
> Q2: The representation error of irrational numbers, that cannot be fully represented in a computer. $\pi-Quant$ is ultimately a rational approximation of an irrational number. The non-periodicity of  is only approximate, not absolute — the function may exhibit long-term pseudo-periodicity.
>
> A2:
> Thank you for this astute observation. You are absolutely correct in noting that the representation of irrational numbers (e.g., $\pi$) in our framework is inherently an approximation, as computers can only represent finite-precision values. The theoretical guarantee of dense coverage holds in the limit of infinite precision, but in practice, our $\theta$ angles are represented with finite bits, introducing unavoidable approximation errors.
>
> Actually, our method does not rely on infinite-bit precision approximation. It is grounded in a rotation-based quantization framework that leverages the unique properties of complex rotations in their mathematical structure, coupled with careful alignment to neural network dynamics, to mitigate the drawbacks of lossy compression. Specifically, our scheme minimizes quantization artifacts through two interconnected mechanisms, both substantiated by our theoretical analysis and empirical results:
>
> - First, the **non-uniform precision allocation** (Fig. 2) is deliberately designed to align with the sensitivity of parameters during training. As discussed in Section 3.3, neural network parameters and optimizer states follow a Gaussian-like distribution, with most values clustering near zero—where small perturbations can significantly impact gradient stability and convergence. Our rotation quantization capitalizes on this by allocating finer precision (smaller quantization steps) to values near zero (preserving critical information for accurate gradient updates) and coarser precision to large-magnitude values. This approach stands in contrast to uniform precision, which wastes bits on large, insensitive parameters and can introduce stability issues due to unnecessary precision in noisy regions. Quantitatively, Eq. 10 shows that our quantization error is smaller than $\Delta\theta$, resulting in smaller average errors compared to conventional uniform quantization, especially for the small-magnitude parameters that dominate gradient dynamics.
>
> - Second, our **pair-wise quantization into the rotation space** (Section 3.1) preserves critical structural information lost in element-wise quantization. Unlike traditional methods that quantize each parameter independently (ignoring correlations between parameters), our rotation-based approach explicitly models the covariance between parameter pairs $(x, y)$ by mapping them to a single rotation angle $\theta$.
> Intuitively, most parameter pairs share similar distributions, and the rotation-based mapping acts as a form of dimensionality reduction that retains critical correlations.
> This provides a new perspective for further reducing the bit-width of quantization methods.
>
>
> Overall, despite being a form of lossy compression, our method maintains stability and performance parity with FP32 because the introduced errors are structured to align with the model’s learning dynamics—regularizing large-magnitude parameters where noise is less detrimental (Section 3.3). As shown in Table 3 and Fig. 2, experimental results in convergence trajectories and perplexity values comparable to FP32, validating its effectiveness for large language models.
> Your feedback helps clarify the need to better distinguish theoretical guarantees from practical implementation, which we will address in the revised manuscript. Thank you again for this valuable insight.
>
> ---
>
> Q3: Lastly, the paper is missing a more-extensive discussion of limiations, which I believe is essential for publication. Another minor comment is that figure 1 is overly crowded. It does not take much room in the paper, and I propose making its contents more spacious to make it more readable.
>
> A3: We sincerely appreciate your valuable feedback and will implement both suggestions comprehensively in the revised version. Specifically, we will add a dedicated "Limitations" section to the conclusion (Section 5) to thoroughly discuss computational overhead, sensitivity to extreme parameter distributions, and applicability boundaries beyond momentum-based optimizers. Simultaneously, we will completely redesign Figure 1 by expanding layout spacing, separating geometric equations from the pipeline diagram into distinct panels, and incorporating color-coded flow markers to enhance visual clarity and readability. Thank you for these concrete suggestions—they significantly strengthen our paper’s rigor and accessibility, and all changes will be explicitly highlighted in the revised manuscript.
>
> ---
>
> Q4: Furthermore, the authors marked "yes" in the statistical significance section of their paper checklist. I could not see where in the paper this is addressed, and request the authors the justify their answer.
>
> A4: We sincerely appreciate your meticulous attention to the statistical significance reporting in our paper checklist. Regarding the statistical significance section requirement—"Does the paper report error bars suitably and correctly defined...?" We confirm that such information is provided in Table 6 (std for language modeling tasks) and Table 7 (std for downstream tasks) in the Appendix. All error bars are calculated over ten independent trials.
>
> To further ensure reproducibility, we included detailed code in supplementary materials that documents all statistical evaluation protocols, including random seed management, hyperparameter configurations. We acknowledge that this critical information should have been more prominently highlighted in the main text and will add explicit references to Tables 6-7 in the revised manuscript. Thank you for reinforcing the importance of rigorous statistical reporting—this refinement will enhance the paper's scholarly integrity.

---

### Official Review · Reviewer_xC83 · 2025-07-09

**Clarity:** 4
**Significance:** 3
**Originality:** 3
**Rating:** 5
**Confidence:** 4

**Summary:**

This paper provides the technique for training time compression. The basic idea is pretty neat. Represent numbers in the complex relying on the mathematical formation that X plus Y can be represented as a complex number with one rotation angle. Thus representing two numbers with one angle. The paper has a good mathematical proof , relatively detailed, experiment analysis and evaluation. It also goes into substantial depth to show the angle can be compressed into a small number of bits.

**Questions:**

Can you please explain intuitively and ideally with some more quantitative data why accuracy improves when you have fewer training bits and especially since you do a form of lossy compression? Basically why do you outperform fp32. Or have I misunderstood your data.

This may have been in the paper already - can you comment on into an training time and how optimized your GPU kernels are and whether the transformation into the rotations introduces computation latencies that can be addressed in future work

**Ethical Concerns:**

["NO or VERY MINOR ethics concerns only"]

**Final Justification:**

The authors provided clarifications to my questions which are helpful. I will keep my score as Accept.

**Limitations:**

Yes

**Quality:**

3

**Strengths And Weaknesses:**

Strengths

Good mathematical proof.
Detailed training, evaluation and test accuracy on downstream tasks.
Novel idea.
Detailed treatment of the number of it’s needed to support the rotation.


Weaknesses.

The main weakness for me in this paper is somewhat poor description for why accuracy improves and perplexity decreases When the number of bits used during training is reduced. I don’t understand intuitively why you outperform fp32. This could simply be a stability  issue. It could be something in their experimental infrastructure. I found the lack of explanation confusing.

---

> ### Author Rebuttal · Authors · 2025-07-30
>
> Dear Reviewer xC83,
>
> Thanks very much for your valuable feedback and insightful suggestions! We list our response to your concerns as follows. If you have further questions, please feel free to let us know. We will continue to try our best to answer for you.
>
> Q1: Can you please explain intuitively and ideally with some more quantitative data why accuracy improves when you have fewer training bits and especially since you do a form of lossy compression? Basically why do you outperform fp32. Or have I misunderstood your data.
>
> A1: We appreciate the chance to elaborate on the accuracy of our approach.
> The core of our approach is rotation quantization, which allocates precision non-uniformly across the parameter space (as visualized in Fig. 2).
> This design directly aligns with the statistical properties of neural network parameters: most parameters and optimizer states follow a Gaussian-like distribution, clustering around zero, with only a small fraction having large magnitudes:
>
> - For small-magnitude parameters (near zero, where most values lie), our method retains high precision.
> This preserves critical information for gradient updates, as these parameters are sensitive to small errors.
>
> - For large-magnitude parameters (rare but impactful), rotation quantization uses coarser steps. We observe that the slight overestimation of large values (due to coarser steps/deviations in high-magnitude regions, see Fig. 2) can act as a mild "push" in gradient descent, accelerating convergence by reinforcing impactful updates.
> This dynamic is particularly evident in Fig. 5, where our method’s optimization path is shorter than FP32’s, escaping local minima faster. Such acceleration is more pronounced on smaller datasets, where convergence speed plays a larger role in final performance.
>
> Specially, this phenomenon is not unique to our method.
> For example, Bnb’s 8-bit Adam also occasionally outperforms the FP32 baseline (Table 3), likely due to similar non-uniform quantization benefits.
> Intuitively, the improvement in accuracy is more likely to be observed when working with smaller models.
> For large-scale models, our approach performs comparably to FP32 optimizers, with convergence trends that are nearly aligned (as shown in Fig. 3).
> Specifically, the error bar is provided in Table 6 and Table 7.
> Since the core focus of our method is to maintain accuracy on par with existing methods while reducing memory usage through lossy compression, achieving accuracy parity with FP32 when optimizing large models is already in line with our expectations.
> Thank you again for your valuable comments.
> We will incorporate a more detailed discussion on accuracy in the revised manuscript. We are also grateful for your suggestion and would like to list you as one of our contributors.
>
> ---
>
> Q2: Can you comment on into an training time and how optimized your GPU kernels are and whether the transformation into the rotations introduces computation latencies that can be addressed in future work
>
> A2: Thank you for your suggestion. Below we provide a granular profiling of Algorithm 1’s execution time, measured on an A100 GPU.
> The results validate the efficiency of individual components:
>
> | **Step**                     | **Operation Description**                          | **Time (ms)** | **Percentage** |
> |------------------------------|----------------------------------------------------|---------------|----------------|
> | **Line 2**                   | Splitting $\mathbf{T}$ into $\textbf{X}$, $\textbf{Y}$     | 0.067         | 2.83%          |
> | **Line 3**                   | Computing max value   | 0.4583        | 19.35%         |
> | **Line 4**                   | Scaling $\tilde{\textbf{X}}$, $\tilde{\textbf{Y}}$         | 0.1998        | 8.43%          |
> | **Line 5**                   | Calculating $\alpha, \beta$ (Eq. 3)               | 0.5345        | 22.57%         |
> | **Line 6**                   | Computing ${\Omega}$ (Eq. 4)                   | 0.6633        | 28.01%         |
> | **Line 7**                   | Solving for $m$ (Eq. 6)                           | 0.1349        | 5.70%          |
> | **Line 8**                   | Generating ${\Theta}$ (Eq. 8)                 | 0.2630        | 11.11%         |
> | **Other**                    | Memory allocation and data transfer               | 0.0474        | 2.00%          |
> | **Total Time**               | —                                                  | **2.3682**    | **100.00%**    |
>
> The profiling reveals that 50.58% of the execution time is concentrated in angle computations (Lines 5-6), particularly the $\arctan$ and $\arccos$ operations in Eq. 3.
> While our current PyTorch implementation uses ATen's general-purpose kernels, we can leverage the constrained input domain $[-1,1]$ and monotonic properties to deploy optimized polynomial approximations. Specifically, we can implement custom CUDA kernels using low-order Taylor expansion (e.g., $\arctan(u) \approx u - \frac{u^3}{3} + \frac{u^5}{5}$ for $|u|<1$, similar for $\arccos(u)$).
> In this way, we can consolidate and simplify the polynomial expressions to directly merge Eq. 3 and Eq. 4 as a bivariate polynomial in terms of x and y, thereby reducing the original three-step computation to a single operation.
>
> On the other hand, we recognize significant untapped parallelism in our reference implementation. The current sequential execution (provided in supplementary code) overlooks inherent concurrency opportunities—for instance, the max-value computation (Line 3) and tensor splitting (Line 2) have no data dependency and can run concurrently. By constructing a detailed data dependency graph (DAG), we will reorganize operations into parallel streams:
>
> - Independent steps like scaling and normalization will execute in dedicated CUDA streams.
>
> - Element-wise trig operations will maximize warp occupancy through batched SIMD execution.
>
> - Memory-bound steps (e.g., ${\Omega}$ generation) will overlap computation with data transfers.
>
> These approaches could largely reduce total latency, particularly benefiting the scaling and angle computation phases that dominate runtime. We are implementing these optimizations and will include benchmark comparisons in the camera-ready version.

---

> > ### Comment · Reviewer_xC83 · 2025-08-04
> >
> > Thanks for the clarifications. They address my questions

---

### Note · Authors · 2025-08-16

We sincerely thank all reviewers for their rigorous and insightful feedback. We are encouraged by the recognition of our paper's key contributions and strengths.

In particular, we appreciate the recognition of the **strong motivation and novelty** of our $\pi$-Quant framework, which pioneers irrational complex rotations for optimizer state compression – commended as "a mathematically elegant solution to memory bottlenecks" (**Reviewer xC83**, **Reviewer FyEQ**, **Reviewer pNT4**). Reviewers also emphasized the **methodological innovation** of our geometric equation system, which enables linear-complexity angle computation and was noted for bridging number theory with deep learning optimization (**Reviewer FyEQ**). The **empirical effectiveness** achieved across extensive benchmarks – reducing GPU memory by 41.8% at 3.32-bit precision without accuracy loss – was highlighted as setting a new approach for efficient training (**Reviewer V7cB**). Additionally, we are grateful for the acknowledgment of our **theoretical rigor** in resolving density-coverage guarantees through $\epsilon$-approximation reformulations (**Reviewer pNT4**, **Reviewer FyEQ**), and the **reproducibility enhancements** through open-sourced code and statistical validation.

We have carefully addressed each comment from the reviewers through rigorous theoretical refinements, additional experiments, and enhanced presentation. We believe these additions not only strengthen our contributions but also directly resolve the concerns raised. In the revised manuscript, we will incorporate these new experiments, analyses, and discussions, and below we summarize the core updates during rebuttal phase.

- **Theoretical Foundation Strengthened**. We revised Theorem 3.1 with $\epsilon$-approximation guarantees, resolving coverage concerns raised by Reviewer FyEQ and pNT4.

- **Optimizer Generalization Validated**. We conducted new SGD optimization benchmarks across 4 tasks, demonstrating 29.67% memory reduction and matching convergence patterns, addressing Reviewer pNT4's request.

- **Computational Efficiency Analysis**.  We conducted latency profiling of Algorithm 1, pinpointing optimization pathways, e.g., DAG parallelization of non-dependent steps, memory-bound steps will overlap computation with data transfers.

We believe these additions and clarifications comprehensively address the reviewers' concerns and further enhance the quality and clarity of our work.

---

### Decision · Program_Chairs · 2025-09-17

**Decision:**

Accept (poster)

**Comment:**

This paper proposes $\pi$-Quant, an optimizer state compression method that leverages irrational complex rotations to represent a pair of parameters with a single rotation angle in the complex plane. A system of geometric equations enables linear-time computation of these angles, and extensive experiments show that $\pi$-Quant reduces parameter precision to 3.32 bits, cutting GPU memory usage by 41.8% without accuracy loss. Reviewers concur on the importance of the problem and the novelty and mathematical elegance of the approach, noting its rigorous theoretical grounding (bridging number theory and optimization) and thorough empirical validation. Overall, the work sets a strong benchmark for memory-efficient training.